# miR-542-3p Attenuates Bone Loss and Marrow Adiposity Following Methotrexate Treatment by Targeting sFRP-1 and Smurf2

**DOI:** 10.3390/ijms222010988

**Published:** 2021-10-12

**Authors:** Ya-Li Zhang, Liang Liu, Yu-Wen Su, Cory J. Xian

**Affiliations:** UniSA Clinical and Health Sciences, University of South Australia, Adelaide, SA 5000, Australia; yali.zhang@mymail.unisa.edu.au (Y.-L.Z.); liang.liu@csl.com.au (L.L.); yu-wen.su@unisa.edu.au (Y.-W.S.)

**Keywords:** methotrexate, miRNA-542-3p, bone formation, marrow adiposity

## Abstract

Intensive methotrexate (MTX) treatment for childhood malignancies decreases osteogenesis but increases adipogenesis from the bone marrow stromal cells (BMSCs), resulting in bone loss and bone marrow adiposity. However, the underlying mechanisms are unclear. While microRNAs (miRNAs) have emerged as bone homeostasis regulators and miR-542-3p was recently shown to regulate osteogenesis in a bone loss context, the role of miR-542-3p in regulating osteogenesis and adipogenesis balance is not clear. Herein, in a rat MTX treatment-induced bone loss model, miR-542-3p was found significantly downregulated during the period of bone loss and marrow adiposity. Following target prediction, network construction, and functional annotation/ enrichment analyses, luciferase assays confirmed sFRP-1 and Smurf2 as the direct targets of miR-542-3p. miRNA-542-3p overexpression suppressed sFRP-1 and Smurf2 expression post-transcriptionally. Using in vitro models, miR-542-3p treatment stimulated osteogenesis but attenuated adipogenesis following MTX treatment. Subsequent signalling analyses revealed that miR-542-3p influences Wnt/β-catenin and TGF-β signalling pathways in osteoblastic cells. Our findings suggest that MTX treatment-induced bone loss and marrow adiposity could be molecularly linked to miR-542-3p pathways. Our results also indicate that miR-542-3p might be a therapeutic target for preserving bone and attenuating marrow fat formation during/after MTX chemotherapy.

## 1. Introduction

Methotrexate (MTX) is known as a useful chemotherapy reagent that is utilised to treat autoimmune diseases and cancers, such as rheumatoid arthritis, head cancer, osteosarcoma, and leukemia, particularly the major childhood cancer acute lymphoblastic leukemia (ALL) [1,2,3,4]. Clinical studies have shown that intensive use of MTX induces skeletal defects, including reduced bone mineral density (BMD) or bone loss, bone growth arrests and marrow adiposity [5,6]. MTX experimental investigations in rats have also revealed the increased adipogenic potential and the decreased osteogenic potential in the bone marrow [7,8]. Specifically, MTX treatment was found to reduce osteogenic differentiation potentials of bone marrow stromal cells (BMSCs) by inhibiting the expression of osteogenesis-related genes, such as runt-related transcription factor 2 (RUNX2) and osterix (OSX) [7,8]. Conversely, adipogenesis was induced following MTX treatment, accompanied by the increased expression of adipogenesis-related genes, such as peroxisome proliferator activated receptor gamma (PPARγ) [7,8]. In an acute intensive MTX treatment model in rats (5 daily administrations at 0.75 mg/kg), altered expression of Wnt/β-catenin signalling pathway components was observed, including increased mRNA expression of Wnt antagonists secreted frizzled related protein 1 (sFRP-1) and dickkopf-1 (DKK1) in rat bones, decreased activation of β-catenin in BMSCs, and reduced mRNA expression of β-catenin target genes lymphoid enhancer factor 1 (LEF1), cyclin D1 and survivin in bones, indicating that MTX treatment attenuated Wnt/β-catenin signalling in bones [7]. Subsequent studies showed that the preservation of Wnt/β-catenin signalling partially abrogated MTX-induced switch in osteogenic and adipogenic commitment [7]. However, despite these previous studies, the molecular mechanisms for this MTX-induced bone/marrow fat homeostasis disorder are unclear, and preventative treatments are lacking.

Recently, microRNAs (miRNAs), non-coding RNAs with 19–25 bp in length, have emerged as critical regulatory molecules involved in BMSC lineage commitment and differentiation [9,10]. Several miRNAs have been reported to manipulate osteogenesis and adipogenesis for maintaining bone/marrow fat balance [11,12,13,14,15]. The miR-542-3p is a well-characterized tumour suppressor gene in cancers and it has been shown to downregulate tumour progression [16,17]. In previous studies, miR-542-3p has also been identified to be involved in regulating osteogenesis. Kureel et al. found that miR-542-3p is a suppressor for osteoblast cell proliferation and differentiation by directly targeting BMP-7 [18]. BMP-7 has exhibited high osteogenic activity by inducing the expression of osteogenesis-related gene markers (e.g., alkaline phosphatase (ALP)) and accelerating calcium deposition for bone matrix mineralization [19]. Inhibition of BMP-7 might impair TGF-β/BMP signalling and further interfere osteoblast differentiation and bone formation. Zhang et al. showed that in ovariectomized (OVX) rat bone loss models, miR-542-3p acts as a positive factor for osteoblast differentiation [20]. miR-542-3p was found to silence the expression of a Wnt signalling inhibitor sFRP-1, which binds to Wnt ligands to inhibit the interaction between the ligands and the receptor complex, thereby, suppressing Wnt/β-catenin signalling for osteogenesis [20]. These studies have suggested that miR-542-3p might be an important regulator involved in bone formation. However, whether miR-542-3p pathways are involved in MTX chemotherapy-induced bone/fat imbalance are unknown.

The current study has investigated potential differential expression of miR-542-3p in bones of MTX-treated rats and performed bioinformatic analyses to construct the miR-542-3p regulatory network. Furthermore, to investigate the role and molecular mechanisms of miR-542-3p in regulating osteogenesis and adipogenesis, by using in vitro models, this study has assessed if the supplementation of miR-542-3p would mitigate MTX-induced reduced osteogenesis and increased adipogenesis and has investigated the associated key pathways (Wnt/β-catenin and TGF-β signalling pathways) being modulated.

## 2. Results

### 2.1. Downregulation of miR-542-3p in Bones of MTX-Treated Rats

To determine if miR-542-3p is differentially expressed in bones of MTX-treated rats, its expression levels were measured using RT-qPCR in tibial bone samples (containing cortical and trabecular bones of the metaphyseal region with bone marrow already having been flushed out) from the MTX time course study (different time points after 5 once-daily MTX injections, *n* = 6/time point). When compared with the saline control, miR-542-3p expression was significantly decreased on days 6 (*p* < 0.001) and 9 (*p* < 0.01) post first injection, two time points previously shown to have significant bone loss and bone marrow adiposity [21]. Expression recovered to pre-treatment level on day 14 (Figure 1A), a time point shown to have bone and bone marrow histological structure recovery [21].

### 2.2. Target Gene Prediction and Network Analyses for miR-542-3p

TargetScan and miRDB were applied to predict the potential targets of miR-542-3p, and its protein–protein interaction (PPI) network was constructed using STRING database and Cytoscape [22,23]. In total, 226 nodes (key genes) and 303 edges were identified in the PPI network complex according to the criteria of “confidence ≥0.4” (Appendix A). Further selection was performed on a degree centrality of ≥4 (Figure 1B). Subsequently, the miR-542-3p-mRNA network was constructed (Figure 1C). As a result, 62 miRNA-mRNA (target gene) pairs were identified, and these mRNAs were considered as key candidate mRNAs for miR-542-3p.

### 2.3. Functional Annotation and Enrichment Analyses of miR-542-3p

The biological functions of key genes of miR-542-3p were defined by Kyoto Encyclopedia of Genes and Genomes (KEGG) pathways and Gene ontology (GO) analyses [24]. KEGG pathways were used to understand high-level biological functions and systems. In total, 11 enriched KEGG pathways were identified (Figure 2A). Among them, alanine, aspartate and glutamate metabolism and GABAergic synapse were identified as significant KEGG pathways (FDR *p* value < 0.05). GO analyses were performed for enrichment analyses on gene sets. The significantly enriched GO terms were shown in Figure 2B (FDR *p* value < 0.05). The GO analyses exhibited that cytoplasm, perinuclear region of cytoplasm, and extracellular exosome were the top three significantly enriched GO terms. The functional annotation and enrichment analyses indicated that the targets genes of miR-542-3p might be involved in various regulations following MTX treatment.

### 2.4. sFRP-1 and Smurf2 as Direct Targets of miR-542-3p

While a previous OVX-induced bone loss model study has demonstrated that sFRP-1 is a direct target of miR-542-3p [20], the target prediction and network analyses in the current study did not identify sFRP-1 as a key target of miR-542-3p. Conversely, these target prediction and network analyses indicated that Smurf2 might be a key target of miR-542-3p. To examine the relevance of sFRP-1 and Smurf2 as potential target genes in the MTX-induced bone defect context, the expression profiles of sFRP-1 (Figure 3A) and Smurf2 (Figure 3B) were analysed by RT-qPCR in the bone samples (containing cortical and trabecular bones at the metaphysis region of tibia) from the MTX time course study. Results showed that the expression of sFRP-1 was elevated on day 3 (*p* < 0.01 compared to saline control) and then declined to the near normal level by day 6. This expression profile showed a negative correlation with that of miR-542-3p. Results of RT-qPCR demonstrated that the mRNA expression of Smurf2 was pronouncedly downregulated throughout the MTX time-course study (from day 3 to day 14, *p* < 0.0005 compared to saline control) (Figure 3B).

To further assess whether sFRP-1 and Smurf2 are the direct targets of miR-542-3p, dual-luciferase assays were performed to determine the potential interactions. As shown in the schematic diagrams, the putative miR-542-3p seed sequence in the 3′ UTR of the sFRP-1 (Figure 3C) or of Smurf2 (Figure 3D) was revealed. As the dual-luciferase assays showed, the luciferase activity was considerably decreased when cells were co-transfected with miR-542-3p agomir (Ago) and pmirGLO-sFRP-1 wild-type (WT) vector (Figure 3C) or pmirGLO-Smurf2 wild-type (WT) vector (Figure 3D). However, no significant changes in luciferase activity were found in cells co-transfected with negative control (NC) or miR-542-3p agomir together with pmirGLO-sFRP-1 mutated (MUT) vector (Figure 3C) or pmirGLO-Smurf2 mutated (MUT) vector (Figure 3D). These results suggested that sFRP-1 and Smurf2 are the direct targets of miR-542-3p.

### 2.5. miR-542-3p Inhibits sFRP-1 and Smurf2 Expression in Osteoblastic Cells

To determine the effects of miR-542-3p treatment on expression of targets sFRP-1 and Smurf2 in osteoblasts, MC3T3.E1 osteoblastic cells treated with negative control (NC) or miR-542-3p agomir for 48 h were harvested for RT-qPCR and Western blot analyses. The mRNA expression levels of sFRP-1 (Figure 4A) and Smurf2 (Figure 4B) in miR-542-3p agomir-treated cells were found not to significantly change when compared to the NC controls. However, the protein expression levels of sFRP-1 (~30 kDa, Figure 4C) and Smurf2 (~93 kDa, Figure 4D) in treated cells were notably decreased, as shown by the Western blots and by the quantitative (normalized) signals of target protein sFRP-1 (Figure 4C) and Smurf2 (Figure 4D). The target protein signals in miR-542-3p treated groups were significantly reduced (sFPR-1, *p* < 0.01 compared to NC; Smurf2, *p* < 0.0005 compared to NC), suggesting that miR-542-3p negatively regulates the expression of sFRP-1 and Smurf2 post-transcriptionally.

### 2.6. miR-542-3p Enhances Osteogenesis and Mineralization after MTX Treatment

To assess effects of miR-542-3p on osteogenesis in MC3T3.E1 osteoblastic cells following MTX treatment, on osteogenic induction day 1, cells were treated with 10 µM of MTX for 48 h, followed by miR-542-3p transfection. Non-treated osteoblastic cells were used as a control. Treatment effects on the osteogenic capacity were examined by a matrix mineralization assay (Figure 5A,B) and RT-qPCR analyses of osteogenesis marker genes on day 21 (Figure 5C). Mineralization of the cells as shown by Alizarin Red S (ARS) staining was significantly lower in MTX+NC treated group compared with non-treated control (*p* < 0.0005, Figure 5A,B), demonstrating MTX treatment-induced inhibitory effect in matrix mineralization. Consistently, expression of osteogenesis marker genes, including RUNX2, ALP, OSX and osteocalcin (OCN), was dramatically declined in MTX+NC treated group when compared to non-treated control (Figure 5C). However, when cells were treated with MTX+miR-542-3p, MTX treatment-induced damaging effect on osteogenic differentiation/mineralization was attenuated or had a trend to be rescued, as demonstrated by enhanced matrix mineralization (*p* < 0.01, Figure 5A,B) and expression of osteogenesis-related genes (for RUNX2, ALP and OSX, *p* < 0.05 or 0.01; for OCN, *p* = 0.09) (Figure 5C). These results indicated that MTX chemotherapy suppresses osteogenesis and mineralization, and its inhibitory effects could be attenuated by miR-542-3p.

### 2.7. miR-542-3p Alleviates Increased Adipogenesis after MTX Treatment

To further examine the effects of miR-542-3p on adipogenesis in 3T3 F442A adipogenic cells following MTX treatment, on adipogenic induction day 1, were treated with 10 µM of MTX for 48 h, followed by miR-542-3p transfection. Non-treated osteoblastic cells were used as a control. Treatment effects on the adipogenic capacity were assessed by an adipogenesis assay (Figure 6A,B) and RT-qPCR analyses of marker genes (Figure 6C) on day 17 of adipogenesis induction. Compared with the non-treated control, a greater proportion of cells treated with MTX+NC was capable of differentiating into mature adipocytes, which characterized by increased Oil Red-O (ORO) positively stained cells and higher ORO absorbance reading at 490 nm (Figure 6A,B). In addition, the expression of two master adipogenic transcription factor genes, CCAAT/enhancer binding protein alpha (C/EBPα) and PPARγ, were significantly increased on day 17 in MTX+NC treated group when compared to non-treated control (Figure 6C). Additionally, the enhancement of adipogenesis by MTX+NC was almost blocked by miR-542-3p treatment, as indicated by declined fat droplet ORO-staining (*p* < 0.01, Figure 6A,B) and downregulated expression of adipogenesis-related genes when compared with the MTX+NC group (C/EBPα *p* < 0.0005 and PPARγ *p* < 0.01, Figure 6C). Moreover, it was found that the effects of MTX+miR-542-3p on adipogenesis was not significantly different when compared with non-treated controls. Together, these findings suggested that MTX treatment promotes adipocyte differentiation and fat accumulation, while miR-542-3p alleviates the increased adipogenesis following MTX treatment.

### 2.8. miR-542-3p Activates Wnt/β-Catenin Signalling in Osteoblastic Cells

The molecule sFRP-1 has been identified as an important Wnt antagonist involved in Wnt/β-catenin signalling [25]. To verify if miR-542-3p influences on Wnt/β-catenin signalling in osteoblastic cells, we examined the mRNA expression of downstream genes of Wnt/β-catenin signalling, namely LEF1 and T-cell factor 7 (TCF7) in MC3T3.E1 cells after miR-542-3p treatment (Figure 7). After miR-542-3p transfection, the mRNA level of LEF1 showed a trend of being elevated (*p* = 0.4204) and that of TCF7 was significantly increased (*p* < 0.05) (Figure 7A). In addition, effect of miR-542-3p on protein expression of β-catenin was assessed by Western blot analyses, and results revealed that phosphorylated β-catenin (~95 kDa, Figure 7B) was decreased (*p* < 0.001), while total active β-catenin (~93 kDa, Figure 7C) was considerably increased (*p* < 0.0005) following miR-542-3p treatment. Collectively, these results suggested that miR-542-3p activates Wnt/β-catenin signalling in osteoblastic cells.

### 2.9. miR-542-3p Inhibits TGF-β Signalling in Osteoblastic Cells

Smurf2 has been characterized as a negative regulator of TGF-β signalling [26]. To determine if miR-542-3p affects Smurf2 expression and then TGF-β signalling in osteogenic cells, the expression of Smad3 (Figure 8A) and RUNX2 (Figure 8B) (identified as direct targets of Smurf2 [27]) was determined by RT-qPCR in MC3T3.E1 cells transfected with or without miR-542-3p. We found that the relative mRNA expression of Smad3 (Figure 8A, *p* < 0.01) and RUNX2 (Figure 8B, *p* < 0.01) was sharply increased after miR-542-3p transfection. To further determine the effects of miR-542-3p on TGF-β signalling, levels of mRNA (Figure 8C) and protein (Figure 8D) expression of downstream gene DLX5 were measured. Results demonstrated that overexpression of miR-542-3p caused a reduction in DLX5 expression at both mRNA and protein levels. The findings implied that miR-542-3p might depress TGF-β signalling in osteoblastic cells.

## 3. Discussion

Intensive application of MTX chemotherapy has been demonstrated to cause increased marrow fat but decreased bone formation (bone/fat switch) in both patients and experimental animals, for which the molecular basis requires more investigations [7,8,28]. Previously, it has been shown that miR-542-3p is associated with various diseases and bone homeostasis [16,17,18,20,29]. The current study has elucidated the potential roles of miR-542-3p in regulating bone/fat formation following MTX treatment. Using a rat MTX intense treatment model (5 daily injections), miR-542-3p was found to be significantly downregulated on day 6 and 9 following the first injection, time points found previously to have bone loss and bone marrow adiposity [21]. Bioinformatic analyses, transfection, luciferase assays and gene/protein expression analyses have shown that sFRP-1 and Smurf2 are the direct targets of miR-542-3p and that miR-542-3p treatment downregulated sFRP-1 and Smurf2 in osteoblastic cells. Additionally, using in vitro models, osteogenesis was enhanced by miR-542-3p in preosteoblastic cells, whereas adipogenesis was attenuated in preadipogenic cells by miR-542-3p following MTX treatment. Furthermore, our results suggest that miR-542-3p in preosteoblastic cells can preserve Wnt/β-catenin signalling but suppress TGF-β signalling following MTX treatment.

### 3.1. miR-542-3p Is Downregulated in Bones of MTX-Treated Rats and sFRP-1 and Smurf2 Are Direct Targets of miR-542-3p

By treating rats with MTX and conducting PCR gene expression analyses, we found the significant downregulation of miR-542-3p in the bone samples during the time points found with bone loss and bone marrow adiposity, suggesting its potential role in mediating the bone/fat switch caused by MTX treatment. To identify its potential target genes and its functions, target gene prediction for miR-542-3p was conducted using online database, and subsequently, the PPI network and miR-542-3p-key mRNA regulatory network were constructed using STRING and Cytoscape. Our bioinformatics analyses of key target genes revealed that alanine, aspartate and glutamate metabolism and GABAergic synapse were significant KEGG pathways and in total, eight significantly enriched GO terms were identified, which might indicate the potential molecular mechanisms of miR-542-3p upon MTX therapy. Inconsistent with our findings, previous studies have shown that miR-542-3p was significantly increased in osteosarcoma tissue, and its bioinformatics analyses revealed that the sphingolipid signalling pathway (*p* = 3.91 × 10^−5^) and bone development were the most highlighted pathway and GO term, respectively [30]. These investigations demonstrated that the target genes and biological functions of miR-542-3p may vary with different pathological conditions.

Previous works have found that miR-542-3p regulates bone formation by targeting sFRP-1 [20]. In addition to confirming the previous findings showing miR-542-3p binding to 3′ UTR of sFRP-1, our current study has also, for the first time, demonstrated that miR-542-3p also targets the 3′ UTR of Smurf2. Furthermore, our results showed that miR-542-3p transfection can significantly downregulate the protein expression of sFRP-1 and Smurf2 in miR-542-3p in osteoblastic cells, which supports previous findings [20].

### 3.2. miR-542-3p Attenuates MTX-Induced Changes in Osteogenesis/Adipogenesis Associated with Alterations in Wnt/b-Catenin Signalling and TGF-b Signalling

The current study has addressed the potential roles of miR-542-p in bone/fat formation by using MC3T3.E1 preosteoblastic and 3T3 F442A preadipocytic cells which were treated with MTX, followed by transfection with miR-542-3p or a negative control. By assessing the treatment effects on osteogenesis and calcified nodules, our findings suggested that miR-542-3p promoted osteoblast differentiation and matrix mineralization, which were consistent with results of a previous study [20]. In addition, the miR-542-3p in combination with MTX treatment rescued the MTX-induced osteogenesis defects, characterized by significantly increased expression of osteogenesis gene markers and calcification of osteoblastic nodules. In addition, by detecting treatment effects on the adipocyte differentiation potentials in 3T3 F442A preadipocytic cells, the remarkable suppression of MTX-induced expression of adipogenic transcription factors PPARγ and C/EBPα and fat droplet accumulation in MTX+miR-542-3p treated group (when compared with MTX+NC treated group) suggested the inhibitory effect of miR-542-3p in MTX-induced adipogenesis. Our study has confirmed that supplementary treatment of miR-542-3p preserves osteoblast differentiation but attenuates adipocyte differentiation in vitro. However, future in vivo studies will be required to address whether supplementary treatment of miR-542-3p can prevent/attenuate MTX-induced bone loss and marrow adiposity.

Previous evidence showed that Wnt signalling and TGF-β signalling are the highlighted signalling pathways that can be affected by miR-542-3p [30,31]. Consistently, sFPR-1 and Smurf2, two direct targets identified in the current study, are known as antagonists for Wnt signalling and TGF-β signalling, respectively. The current study showed that miR-542-3p inhibits the expression of both sFPR-1 and Smurf2 at the post-transcriptional level. As steps to further understand the underlying mechanisms by which miR-542-3p regulates bone/fat formation, we have investigated the effects of miR-542-3p transfection on activation of Wnt signalling and TGF-β signalling pathways in osteoblastic cells. For the canonical Wnt/β-catenin signalling pathway, a Wnt ligand binds to the receptors, preventing the phosphorylation of β-catenin by disrupting the destruction complex, causing β-catenin translocation into the nucleus and interaction with the T-cell factor/lymphoid enhancer factor (TCF/LEF), triggering target gene expression [32]. Using RT-qPCR analyses and Western blot studies, we found enhanced expression of LEF1 and TCF7, lowered expression of phosphorylated β-catenin and increased expression of active β-catenin in miR-542-3p-treated osteoblastic cells, indicating increased activation of Wnt/β-catenin signalling in osteoblastic cells after miR-542-3p treatment. These findings are consistent with the finding of miR-542-3p targeting sFRP1 [20], which is a major antagonist of Wnt/β-catenin signalling.

For TGF-β/Smad signalling, a TGF-β ligand binds to the TGF-β receptor complex which can trigger Smad2/3-dependent signalling pathways, and Sumrf2 has been identified to disrupt the Smad2/3 and negatively regulates TGF-β/Smad signalling [26]. In our present work, we demonstrated that miR-542-3p transfection increased mRNA expression of Smad3 and RUNX2, which are direct targets of Smurf2, suggesting that miR-542-3p might function in regulating osteogenesis also through modulating TGF-β/Smad signalling, which is consistent with a previous study [33]. Furthermore, miR-542-3p treatment was found to sharply reduce the expression of downstream transcriptional factor DLX5 at both mRNA and protein levels in osteoblastic cells. This finding is somewhat surprising; as the antagonist Smurf2 expression was decreased by miR-542-3p treatment, the expression of DLX5 would be expected to be induced for TGF-β signalling activation. As a potential explanation, it is possible that miR-542-3p may also target other molecules which contribute to TGF-β signalling inhibition. Previous work found that miR-542-3p directly targets BMP-7, TGF-β1, integrin-linked kinase (ILK) and Smad2, thereby, suppressing the TGF-β signalling pathway [16,18,31,34]. Further studies are required to investigate whether and how miR-542-3p modulates Wnt/β-catenin signalling and TGF-β signalling pathways, and thus contributes to observed bone/fat formation regulation.

## 4. Materials and Methods

### 4.1. MTX Treatment in Rats

MTX at 0.75 mg/kg was once daily injected subcutaneously into groups of six-week-old male Sprague-Dawley rats (~120 g) for five consecutive days. The dosage applied is relevant to the therapeutic usage in paediatric oncology [35,36,37]. To observe MTX treatment effects on miR-542-3p expression in bones in a time course, rats were sacrificed by CO_2_ overdose on day 3, 6, 9 and 14 following the first injection (*n* = 6/time point) as previously described [21,35]. Six rats in untreated control received saline and were sacrificed on day 14 [35]. All animal procedures were approved by the Animal Ethics Committee of the University of South Australia. The metaphyseal region of tibia was collected and stored at −80 °C for gene expression analyses.

### 4.2. RNA Isolation and RT-qPCR

RNA was extracted using mirVana™ miRNA isolation kit (Thermo Fisher Scientific, Scoresby, VIC, Australia) from the collected frozen tibial metaphyseal bone samples (containing cortical and trabecular bone without bone marrow). In addition, total RNA from cultured cells (MC3T3.E1 cells and 3T3 F442A cells) was isolated and purified using the GenElute^TM^ Total RNA Purification Kit (RNB100-100RXN, Sigma-Aldrich, North Ryde, NSW, Australia). First-strand cDNA was then reverse transcribed using iScript™ Advanced/Selected cDNA Synthesis Kit for RT-PCR (Bio-Rad Laboratories, Gladesville, NSW, Australia). Specific Primers (Table 1) were designed using NCBI Primer Blast and Primer 3.0 InPut software and synthesized by Sigma-Aldrich. To quantity the genes of interest, quantitative PCR was performed using SsoAdvanced™ Universal SYBR^®^ Green Supermix (Bio-Rad) and run-in triplicate on a CFX PCR system (Bio-Rad). Relative expression of genes was calculated using the comparative 2^−ΔΔCt^ method normalized to the expression of internal housekeeping gene cyclophilin A (CycA) or U6.

### 4.3. Prediction of Target Genes for MiR-542-3p

The potential targets of miR-542-3p were predicted using online database miRDB (http://mirdb.org/index.html, accessed on 25 September 2020) and TargetScan (http://www.targetscan.org/vert_72/, accessed on 25 September 2020) as we recently described [38].

### 4.4. Network Construction

STRING (http://string-db.org, accessed on 8 May 2021) is an online biological database that aims to provide the evaluation and integration of protein-protein interactions (PPI) [39]. In the current study, STRING was used to build a PPI network associated with miR-542-3p. The required interaction score was set to 0.4 confidence and the organism to Rattus norvegicus, and Cytoscape 3.8.2 (https://cytoscape.org/, accessed on 8 May 2021) was then used to visualize and analyse the network [22,23]. Key genes were identified in the network with a degree centrality of ≥4 set as the criterion. Subsequently, the miRNA–mRNA gene network was constructed using Cytoscape as we recently described [38].

### 4.5. Functional Annotation and Enrichment Analyses

An online program called Database Annotation for Visualization and Integrated Discover (DAVID) was applied to miRNA candidates obtained [24], which allows the functional analyses of genes obtained from various genomic resources. Gene ontology (GO) and Kyoto Encyclopedia of Genes and Genomes (KEGG) pathway analyses were performed to implement the enrichment analyses as we recently described [38]. False discovery rate (FDR) *p* values < 0.05 were considered as significantly enriched pathways and GO terms.

### 4.6. Cell Culture and In Vitro Osteoblast Differentiation

MC3T3.E1 osteoblastic precursor cells (CellBank Australia, Westmead, NSW, Australia) were seeded in groups of 6-well plates/24-well plates with basal medium containing α-MEM (Sigma), 10% FBS (Invitrogen, Carlsbad, ON, Canada), 15 mM HEPES (Thermo Fisher Scientific), 1% antibiotic-antimycotic (Thermo Fisher Scientific) and 2 mM L-glutamine (Sigma) at 37 °C with 5% CO_2_ until 70–80% confluence. To induce osteogenesis, confluent cells were cultured with basal medium supplemented with 10 nM dexamethasone (Sigma) and 10 mM β-glycerol-2-phosphate (Sigma). The miR-542-3p agomir or negative control (BioNovus Life Sciences, Cherrybrook, NSW, Australia) was transfected into cells by using Lipofectamine^®^ 2000 reagent (Thermo Fisher Scientific). The medium was refreshed every 2 days until further experiments.

### 4.7. Alizarin Red S (ARS) Staining for Mineralization

On days 21–23, osteogenically differentiated cells were fixed by 10% formalin (Sigma) and then stained by 40 mM Alizarin Red S (ARS, Sigma) at room temperature for 30 min with gentle shaking. After being washed with diH_2_O, cells were observed and photographed using a microscope. Then, cells were incubated with 10% acetic acid (Sigma) for 30 min with shaking. Cells were transferred into a 1.5 mL microcentrifuge tube, vortexed for 30 s. After that, tubes were heated at 85 °C for 10 min and subsequently, incubated on ice for 5 min. The slurry was centrifuged at 20,000× *g* for 15 min and supernatant (~200 µL) was collected into a new tube. To neutralize the acid, 75 µL of 10% ammonium hydroxide (M&B, Ivanhoe, VIC, Australia) was added and 150 µL of samples were prepared in triplicates in 96-well plates. Absorbance was read at 405 nm with a plate reader (PerkinElmer, Melbourne, VIC, Australia).

### 4.8. Cell Culture and In Vitro Adipocyte Differentiation

Preadipocytes 3T3 F442A cells (CellBank Australia) were seeded in groups of 6-well plates/24-well plates with culture medium containing DMEM (Sigma), 2 mM L-glutamine and 10% newborn calf serum (NBCS, Sigma) at 37 °C with 5% CO_2_ until 70–80% confluence. For experiments, confluent cells were grown with a culture medium supplemented with 5 mg/mL insulin (Sigma). The miR-542-3p agomir or negative control was transfected into cells by using Lipofectamine^®^ 2000 reagent. The medium was refreshed every 2 days until further experiments.

### 4.9. Oil Red O (ORO) Staining for Adipogenesis

On days 14–17, adipogenically differentiated cells were fixed with 10% formalin and then incubated with 60% isopropyl alcohol (Sigma) for 5 min. After removing the isopropyl alcohol, cells were stained with Oil Red O (ORO, Sigma) for 30 min with gentle shaking. Cells were washed with diH_2_O to remove the dye, and then hematoxylin (Sigma) was added into each well for 1 min. Then, cells were washed again with diH_2_O before being photographed under a microscope. Then, after washing with 60% isopropyl alcohol 3 times, ORO stain was extracted with 100% isopropyl alcohol for 5 min with gentle rocking, and 150 µL of samples were prepared in triplicates in 96-well plates. Absorbance was read at 490 nm with a plate reader (PerkinElmer).

### 4.10. Dual-Luciferase Assays

The length of 3′ UTR with the potential binding sequence (wild-type, WT) or mutated sequence (MUT) of the potential target gene (sFRP-1 or Smurf2) mRNA was constructed into pmirGLO Dual-Luciferase miRNA Target Expression Vector (Promega, Madison, WI, USA). Plasmid constructs were co-transfected with an agomir or negative control of miR-542-3p into 143B reporter cells (ATCC^®^ CRL-8303™, Invitro Technologies, Noble Park, VIC, Australia) by using Lipofectamine^®^ 2000 reagent. After 24 h, cells were harvested for a dual luciferase assay in triplicate which was performed following the manufacturer’s instructions (Dual-Glo^®^ Luciferase Assay System, Promega). The ratio of luminescence from the Firefly reporter to luminescence from the control *Renilla* reporter was calculated, which was normalized to the ratio of control well (vector only) as we recently described [38].

### 4.11. Western Blot Analyses

RIPA lysis buffer (Thermo Fisher Scientific) containing 10 µL/mL of protease and phosphatase inhibitor cocktail (Thermo Fisher Scientific) was used for protein extraction. To determine the concentrations of protein in samples, the BCA protein assay was performed using the assay kit (Thermo Fisher Scientific). About 8 µg of cell lysate sample protein mixed with laemmli sample loading buffer (Bio-Rad) was loaded onto 4–20% SDS-PAGE protein gels (Bio-Rad). Electrophoresis was conducted at a constant voltage of 90 V for 10 min and then 120 V for about 40 min with Novex™ Tris-Glycine SDS running buffer (Thermo Fisher Scientific). Protein was then transferred to 0.2 µm nitrocellulose transfer packs (Bio-Rad). After the transfer, the membrane was stained with Revert™ 700 total protein stain following the manufacturer’s instructions (LI-COR, Mulgrave, VIC, Australia). The membrane was imaged in the 700 nm channel with a CLx Odyssey^®^ imaging system (LI-COR). After being blocked with 3% milk/TBS blocking buffer at room temperature for 1 h, the membrane was incubated overnight at 4 °C with a specific primary antibody (sFRP-1 polyclonal antibody, PA5-67713, Thermo Fisher Scientific; Smurf2 polyclonal antibody, PA5-114485, Thermo Fisher Scientific; Phospho-β-catenin (Ser33/37/Thr41) antibody, 9561S, Cell Signalling Technologies; Non-phospho (Active)-β-catenin (Ser33/37/Thr41), 8814S, Cell Signalling Technologies; DLX5 polyclonal antibody, PA5-101134, Thermo Fisher Scientific). After washes, the membrane was further incubated with the IRDye^®^ 800CW Donkey anti-rabbit IgG secondary antibody (LI-COR) solution. The immunodetection was visualized in an 800 nm channel with a CLx Odyssey^®^ imaging system.

### 4.12. Statistical Analysis

Image Studio Lite Ver 5.2 (LI-COR) was utilized to quantify the protein expression. Experimental data were analysed by GraphPad Prism 8 (San Diego, CA, USA). Statistical significance was performed via *t* test, one-way or two-way ANOVA followed by Tukey’s or Sidak’s post-test (details were mentioned in legends under each figure). Experiments were performed at least three times. *p* < 0.05 was considered statistically significant. In figures, significant values are marked * *p* < 0.05, ** *p* < 0.01, *** *p* < 0.001, **** *p* < 0.0005. *p* refers to comparison between each treated sample to the non-treated control. Results were displayed as mean ± SEM (*n* = 3).

## 5. Conclusions

This study has shown that the MTX chemotherapy decreases osteogenesis but increases adipogenesis (bone/fat switch) in the bone marrow, which may be associated with reduction in miR-542-3p expression (Figure 9). The current study has also identified Wnt signalling antagonist sFPR-1 and TGF-β signalling antagonist Smurf2 as the direct targets of miR-542-3p. Our results suggest that supplementation of miR-542-3p may attenuate MTX-induced impairment in bone formation and increase in bone marrow fat formation via targeting sFRP-1 and Smurf2, which results in preserving Wnt/β-catenin signalling and suppressing TGF-β signalling following MTX treatment. Collectively, this study has demonstrated a vital role of miR-542-3p in regulating osteogenesis and adipogenesis following MTX treatment and thus has shed lights on the underlying mechanisms of MTX therapy-associated bone/fat switch. From results of our study, it can be proposed that the reduction in miR-542-3p expression in bone is correlated with MTX-induced reduced osteogenesis and increased adipogenesis. Further in vivo studies are required to confirm whether miR-542-3p might be a therapeutic target to control MTX-induced bone/fat switch.

## Figures and Tables

**Figure 1 ijms-22-10988-f001:**
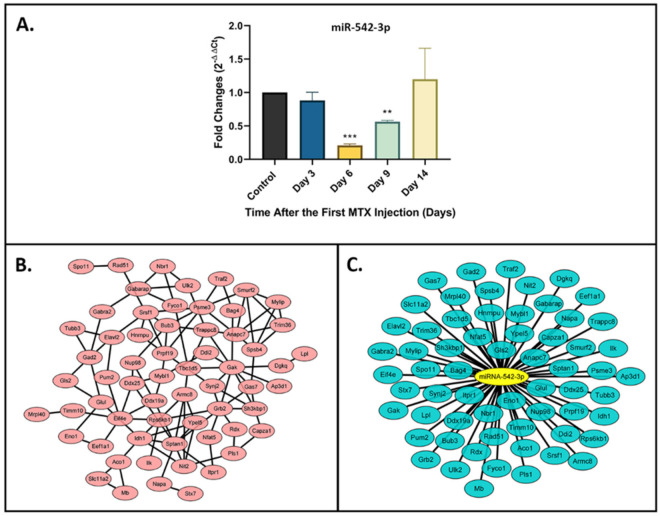
Expression and network analyses of miR-542-3p. (**A**) Expression of miR-542-3p in bone samples (containing cortical and trabecular bones at the metaphyseal region of tibia) from the MTX time course study (*n* = 6/time point). The miR-542-3p expression was significantly decreased on days 6 and 9 post first injection and its expression recovered to pre-treatment level on day 14. Statistical significance was performed via one-way ANOVA followed by Dunnett’s post-test. Significant values are marked ** *p* < 0.01 and *** *p* < 0.001. (**B**) The PPI network of the predicted key genes of miR-542-3p. Using STRING online database and Cytoscape, a total of 226 nodes was filtered (required interaction score was set to 0.4 confidence and the organism to Rattus norvegicus). (**C**) The miRNA-key mRNA network of miR-542-3p. Key genes in the network were identified with a degree centrality of ≥4 set as the criterion.

**Figure 2 ijms-22-10988-f002:**
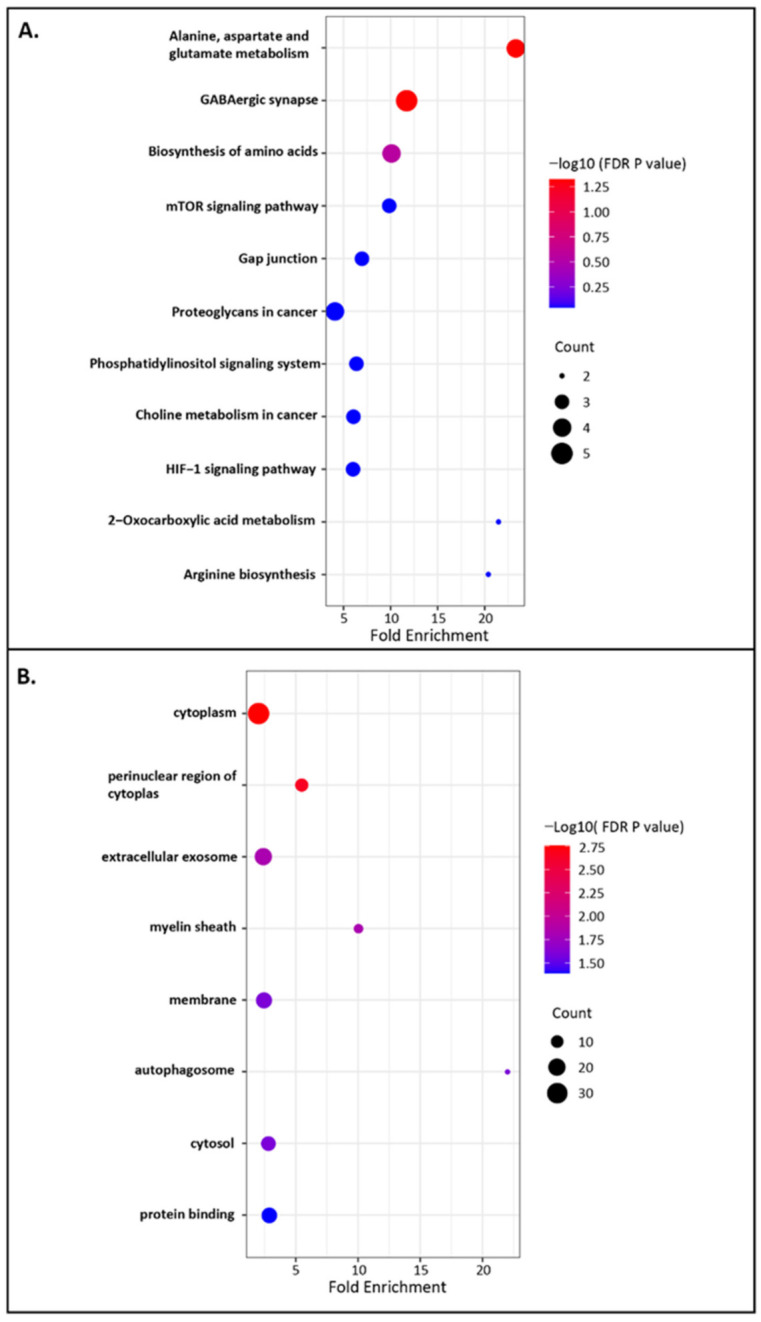
Functional annotation and enrichment analyses of miR-542-3p. (**A**) KEGG pathway analyses of key genes, with a total of 11 enriched KEGG pathways being identified from targets genes, including alanine, aspartate and glutamate metabolism and GABAergic synapse being identified as significant KEGG pathways (FDR *p* value < 0.05). (**B**) GO enrichment analyses of key genes, showing the significantly enriched GO terms (FDR *p* value < 0.05).

**Figure 3 ijms-22-10988-f003:**
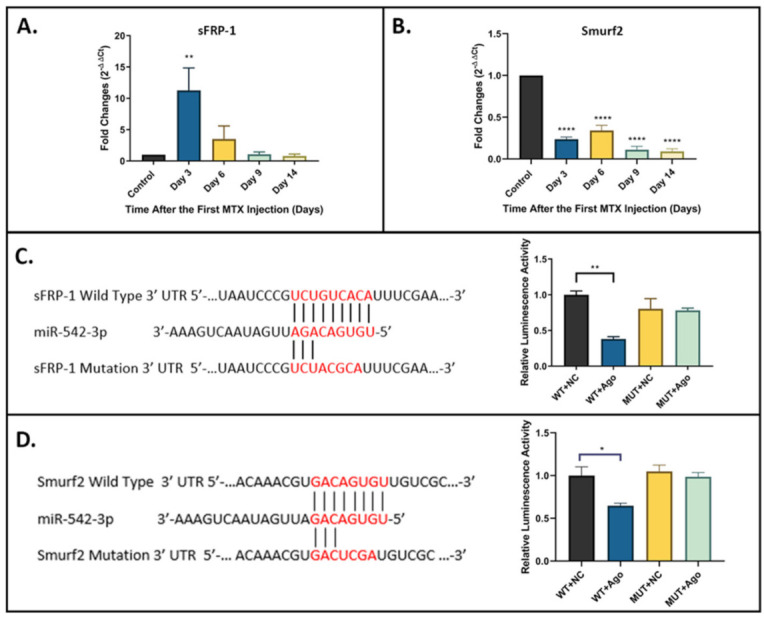
miR-542-3p directly targets sFRP-1 and Smurf2. (**A**) RT-qPCR analyses of sFRP-1 mRNA expression in bone samples from the MTX time course study (*n* = 6/time point). The expression of sFRP-1 was elevated on day 3 and then declined to the near normal level by day 6. (**B**) RT-qPCR analyses of Smurf2 mRNA expression in bone samples. The expression of Smurf2 was pronouncedly downregulated throughout the MTX time-course study. (**C**) (**left**) Schematic diagram of the putative miR-542-3p seed sequence in the 3′ UTR of the sFRP-1; (**right**) relative luminescence activity in the dual luciferase reporter assay. Cells co-transfected with miR-542-3p agomir and pmirGLO-sFRP-1 wild-type vector have shown considerably decreased luciferase activity. (**D**) (**left**) Schematic diagram of the putative miR-542-3p seed sequence in the 3′ UTR of the Smurf2; (**right**) relative luminescence activity in the dual luciferase reporter assay. Cells co-transfected with miR-542-3p agomir and pmirGLO-Smurf2 wild-type vector have shown considerably decreased luciferase activity. Statistical significance analysis was performed via one-way ANOVA followed by Dunnett’s post-test. Significant values are marked * *p* < 0.05, ** *p* < 0.01, and **** *p* < 0.0005. WT: vector with wild-type sFRP-1/Smurf2 seed region; NC: negative control; Ago: miR-542-3p agomir; MUT: vector with mutated sFRP-1/Smurf2 seed region.

**Figure 4 ijms-22-10988-f004:**
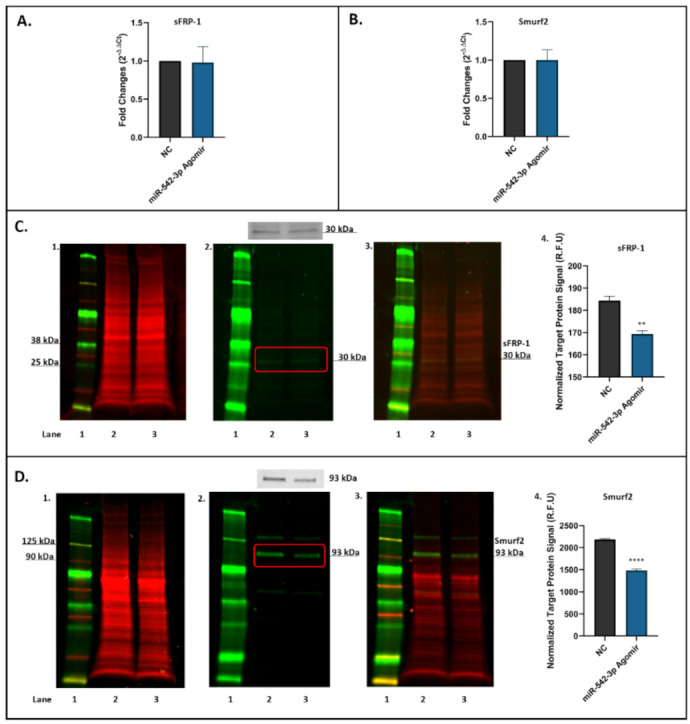
miR-542-3p Inhibits sFRP-1 and Smurf2 expression at the post-transcriptional level in MC3T3.E1 osteoblastic cells. (**A**) RT-qPCR results showed that there were no significant changes in sFRP-1 mRNA expression after miRNA-542-3p transfection when compared to the negative control (NC) group. (**B**) RT-qPCR analyses of Smurf2 mRNA expression after miRNA-542-3p transfection, which was not significantly changed compared to the NC. (**C**) Western blot studies of sFPR-1 after miRNA-542-3p transfection. The protein expression levels of sFRP-1 (~30 kDa) in treated cells were notably decreased compared with NC. (**C1**). Total protein extract was visualized (700 nm channel, red); (**C2**). Target protein sFRP-1 was visualized (800 nm channel, green); (**C3**). Merged images of blots; and (**C4**). Normalized target sFRP-1 protein signal (from 3 experiments). (**D**) Western blot studies of Smurf2 after miRNA-542-3p transfection. The protein expression levels of Smurf2 (~93 kDa) in treated cells were notably decreased compared with NC. (**D1**). Total protein extract was visualized (700 nm channel, red); (**D2**). Target protein Smurf2 was visualized (800 nm channel, green); (**D3**). Merged images of blots; and (**D4**). Normalized target Smurf2 protein signal (from 3 experiments). Treatments on each lane: lane 1: pre-stained protein ladder; lane 2: negative control (NC); lane 3: miR-542-3p agomir. R.F.U: relative fluorescence units. Statistical significance analyses were performed via *t* test. Significant values are marked ** *p* < 0.01 and **** *p* < 0.0005.

**Figure 5 ijms-22-10988-f005:**
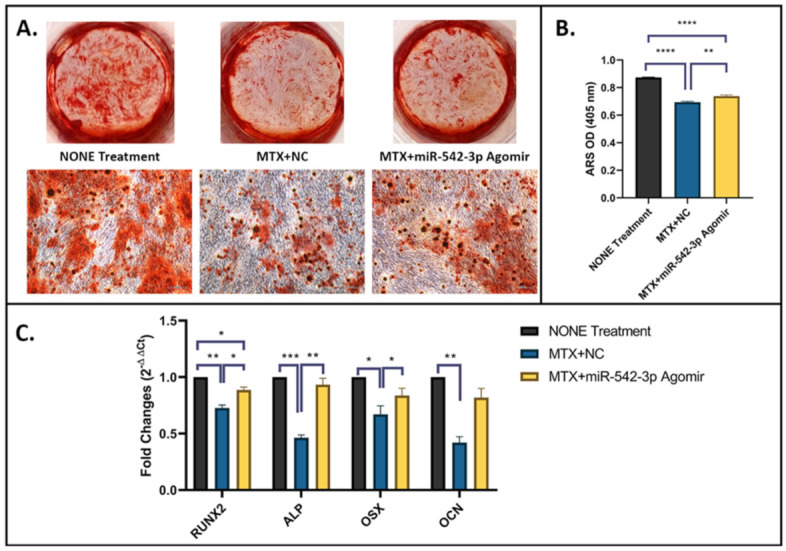
miR-542-3p enhances osteogenesis after MTX Treatment. On osteogenic induction day 1, MC3T3.E1 cells were treated with 10 µM of MTX for 48 h, followed by transfection with miR-542-3p or negative control (NC). Non-treated cells were used as a control. (**A**) Representative images of Alizarin Red S (ARS) staining as performed on day 21 of osteogenic differentiation. Scale bar = 200 µm. (**B**) ARS quantification, which showed a significant reduction in MTX+NC treated group compared with non-treated control, and a partial rescue in cells treated with MTX+miR-542-3p when compared with MTX+NC treated cells. Statistical significance analysis was performed via one-way ANOVA followed by Tukey’s post-test. (**C**) RT-qPCR analyses of mRNA expression of osteogenesis-related marker genes, which showed a significant decline in MTX+NC treated group when compared to non-treated control and a significant rescue (for RUNX2, ALP and OSX, *p* < 0.05 or 0.01) or a trend of rescue (OCN, *p* < 0.09) in cells treated with MTX+miR-542-3p when compared with MTX+NC treated cells. Statistical significance analysis was performed via two-way ANOVA followed by Tukey’s post-test. Significant values are marked * *p* < 0.05, ** *p* < 0.01, *** *p* < 0.001, and **** *p* < 0.0005.

**Figure 6 ijms-22-10988-f006:**
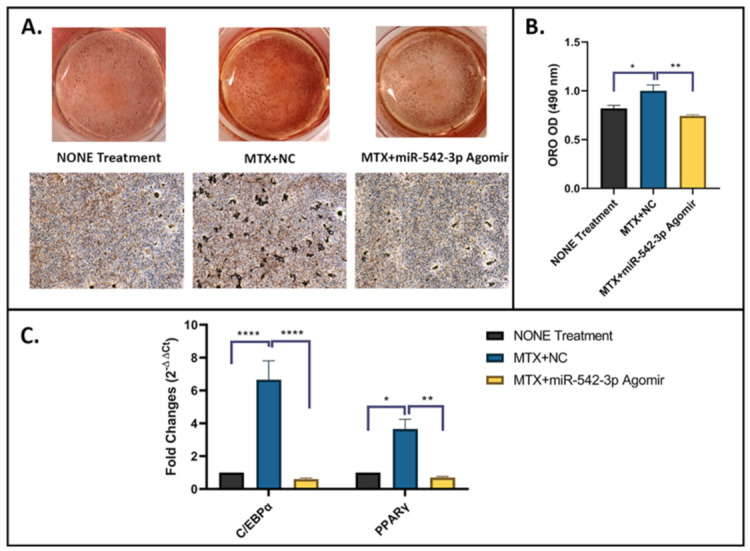
miR-542-3p alleviates MTX-induced increased adipogenesis. On adipogenic induction day 1, 3T3 F442A adipogenic cells were treated with 10 µM of MTX for 48 h, followed by transfection with miR-542-3p or negative control (NC). Non-treated adipogenic cells were used as a control. (**A**) Representative images of ORO staining, which was performed on day 17 of adipogenic differentiation. Scale bar = 200 µm. (**B**) ORO quantification, which showed, compared with non-treated control, a greater oil droplet staining of mature adipocytes observed in cells treated with MTX+NC. The enhancement of adipogenesis by MTX+NC was almost blocked by miR-542-3p treatment. Statistical significance analysis was performed via one-way ANOVA followed by Tukey’s post-test. (**C**) RT-qPCR analyses of C/EBPα and PPARγ on day 17 of adipogenesis. The expression of C/EBPα and PPARγ, was significantly increased on day 17 in MTX+NC treated group when compared to the non-treatment control. Compared with MTX+NC treated group, low expression levels of adipogenesis-related genes were found in MTX+miR-542-3p treated group. Statistical significance analysis was performed via two-way ANOVA followed by Tukey’s post-test. Significant values are marked * *p* < 0.05, ** *p* < 0.01 and **** *p* < 0.0005.

**Figure 7 ijms-22-10988-f007:**
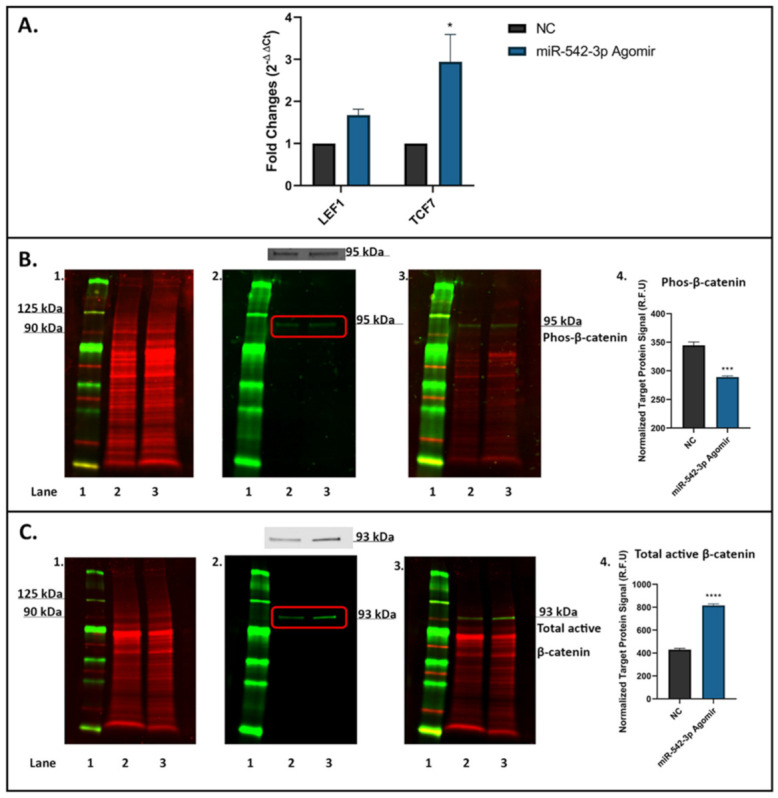
miR-542-3p Activates Wnt/β-catenin signalling in osteoblastic cells. MC3T3.E1 cells were treated with negative control (NC) or miR-542-3p for 48 h. (**A**) RT-qPCR analyses of LEF1 and TCF7 mRNA expression after miRNA transfection. The mRNA level of LEF1 showed a trend of being elevated and that of TCF7 was significantly increased after miR-542-3p treatment. Statistical significance analysis was performed via two-way ANOVA followed by Sidak’s post-test. (**B**) Western blot studies of phosphorylated (Phos) β-catenin after miRNA transfection. The protein expression of Phos-β-catenin (~95 kDa) was decreased after miR-542-3p treatment. (**B1**). Total protein extract was visualized (700 nm channel, red). (**B2**). Target protein phosphorylated β-catenin was visualized (800 nm channel, green). (**B3**). Merged images of blots. (**B4**). Normalized target Phos-β-catenin protein levels (from 3 experiments). Statistical significance analysis was performed via *t*-test. (**C**) Western blot studies of total active β-catenin after miRNA transfection. The protein level of total active β-catenin (~93 kDa) was considerably increased after miR-542-3p treatment. (**C1**). Total protein extract was visualized (700 nm channel, red). (**C2**). Total active β-catenin was visualized (800 nm channel, green). (**C3**). Merged images of blots. (**C4**). Normalized target total active β-catenin protein levels (from 3 experiments). Statistical significance analysis was performed via *t* test. Treatments on each lane: lane 1: pre-stained protein ladder; lane 2: NC; lane 3: miR-542-3p agomir. R.F.U: Relative Fluorescence Units. Significant values are marked * *p* < 0.05, *** *p* < 0.001 and **** *p* < 0.0005.

**Figure 8 ijms-22-10988-f008:**
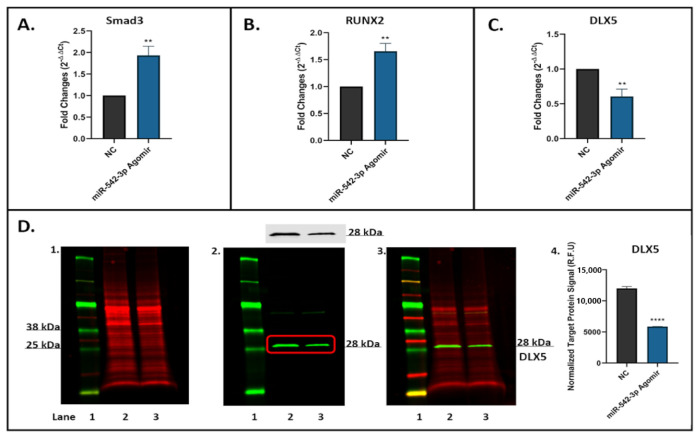
miR-542-3p Inhibits TGF-β Signalling in Osteoblastic Cells. MC3T3.E1 cells were treated with negative control (NC) or miR-542-3p for 48 h. (**A**–**C**) RT-qPCR analyses of Smad3, RUNX2 and DLX5 after miRNA transfection. The expression of Smad3 and RUNX2 was sharply increased after miR-542-3p transfection, while that of DLX5 was significantly declined. (**D**) Western blot studies of DLX5 protein expression after miRNA transfection, which showed a reduction in DLX5 levels after miR-542-3p treatment. (**D1**). Total protein extract was visualized (700 nm channel, red). (**D2**). Target protein DLX5 was visualized (800 nm channel, green). (**D3**). Merged images of blots. (**D4**). Normalized target DLX5 protein levels (from 3 experiments). Treatments on each lane: lane 1: pre-stained protein ladder; lane 2: NC; lane 3: miR-542-3p agomir. R.F.U: Relative Fluorescence Units. Statistical significance analysis was performed via *t* test. Significant values were marked ** *p* < 0.01 and **** *p* < 0.0005.

**Figure 9 ijms-22-10988-f009:**
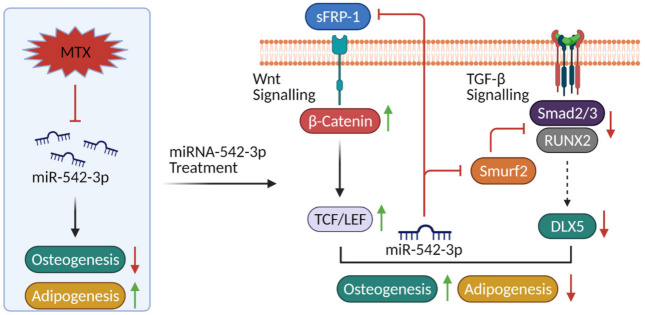
Schematic representation of the role and as a potential therapeutic target for miR-542-3p in MTX treatment-induced bone/fat switch. Methotrexate (MTX) chemotherapy decreases osteogenesis but increases adipogenesis (bone/fat switch) in the bone marrow, which may be associated with reduction in miR-542-3p expression. The secreted frizzled related protein 1 (sFPR-1) and SMAD specific E3 ubiquitin protein ligase 2 (Smurf2) as direct targets of miR-542-3p identified in the current study are known as antagonists for the Wnt/β-catenin signalling and TGF-β signalling. Treatment of miR-542-3p in preosteoblasts and preadipocytes after MTX treatment can attenuate MTX-induced bone/fat switch, which might be linked with preservation of Wnt/β-catenin signalling and suppression of TGF-β signalling.

**Table 1 ijms-22-10988-t001:** Primer Sequences used for RT-qPCR.

Gene	RT Primer	Forward Primer (5′-3′)	Reverse Primer (5′-3′)
rSmurf2	Random Primer (Bio-Rad)	AACACCCGGTTAAAGCACTG	AAACAAGTGTGGGCTTTTGG
mSmurf2	Random Primer (Bio-Rad)	GTGAAGAGCTCGGTCCTTTG	TCGCTTGTATCTTGGCACTG
rsFRP-1	Random Primer (Bio-Rad)	CCCGAGATGCTCAAATGTGAC	AGATGTTCGATGATGGCCTCC
msFRP-1	Random Primer (Bio-Rad)	TGCGAGCCGGTCATGCAGTT	ACACGGTTGTACCTTGGGGCT
mSmad3	Random Primer (Bio-Rad)	CTGGGCCTACTGTCCAATGT	GCAGCAAATTCCTGGTTGTT
mRUNX2	Random Primer (Bio-Rad)	CCCAGCCACCTTTACCTACA	TATGGAGTGCTGCTGGTCTG
mALP	Random Primer (Bio-Rad)	GCTGATCATTCCCACGTTTT	CTGGGCCTGGTAGTTGTTGT
mOSX	Random Primer (Bio-Rad)	ACTCATCCCTATGGCTCGTG	GGTAGGGAGCTGGGTTAAGG
mOCN	Random Primer (Bio-Rad)	AAGCAGGAGGGCAATAAGGT	TTTGTAGGCGGTCTTCAAGC
mC/EBPα	Random Primer (Bio-Rad)	TGGACAAGAACAGCAACGAG	CCTTGACCAAGGAGCTCTCA
mPPARγ	Random Primer (Bio-Rad)	TTTTCAAGGGTGCCAGTTTC	AATCCTTGGCCCTCTGAGAT
mLEF1	Random Primer (Bio-Rad)	TATGAACAGCGACCCGTACA	ACATCTGACGGGATGTGTGA
mTCF7	Random Primer (Bio-Rad)	GCCAGAAGCAAGGAGTTCAC	TACACCAGATCCCAGCATCA
mDLX5	Random Primer (Bio-Rad)	CCACCAGCCAGCCAGAGAAA	GGGGCATCTCCCCGTTTTT
rmCycA	Random Primer (Bio-Rad)	CGTTGGATGGCAAGCATGTG	TGCTGGTCTTGCCATTCCTG
miR-542-3p	GAAAGAAGGCGAGGAGCAGATCGAGGAAGAAGACGGAAGAATGTGCGTCTCGCCTTCTTTCTTCAGTTA	TGTGACAGATTGATAACTGA	GAGGAAGAAGACGGAAGAAT
U6	CGCTTCACGAATTTGCGTG	GCTTCGGCAGCACATATAC	CGCTTCACGAATTTGCGTG

r-primers for rat samples; m-primers for mouse samples; rm-primers for rat and mouse samples. Smad3: SMAD family member 3; Smurf2: SMAD specific E3 ubiquitin protein ligase 2; sFRP-1: secreted frizzled related protein 1; RUNX2: runt-related transcription factor 2; ALP: alkaline phosphatase; OSX: osterix; OCN: osteocalcin; C/EBPα: CCAAT/enhancer binding protein alpha; PPARγ: peroxisome proliferator activated receptor gamma; LEF1: lymphoid enhancer binding factor 1; TCF7: transcription factor 7; DLX5: distal-less homeobox 5; CycA: cyclophilin A.

## Data Availability

The data that support the findings of this study are available on request from the corresponding author.

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
