# Peer review of "miR-542-3p Attenuates Bone Loss and Marrow Adiposity Following Methotrexate Treatment by Targeting sFRP-1 and Smurf2"

_ijms, 2021, doi:10.3390/ijms222010988_

Round 1

Author Response

We highly appreciated your expert insightful and constructive comments and suggestions for this work. The manuscript has been revised carefully according to the comments and suggestions. Changes are shown as red color texts in the revised manuscript. Please find below our responses to your specific comments:

  1. The manuscript requires English-language correction.

Response: Thank you for your suggestion. We have now proofread the manuscript carefully and thoroughly again and have corrected any English language errors that we could find.

  1. Line 44: there is no such thing as calcium mineralization, but there is calcium deposition which leads to bone matrix mineralization.

Response: We appreciated your comments and we apologized for some unclear statements. We have now revised the sentences in the revised manuscript.

  1. Authors in methods section and results section write about bone samples: these were hard bone (compact bone) samples or bone marrow samples or hard bone with bone marrow samples? Please, specify this matter.

Response: We appreciated your suggestions. Bone samples in this manuscript referred to samples containing cortical and trabecular bones at the metaphyseal region of tibia, with the bone marrow already having been flushed out. We have now clarified this in the revised manuscript (methods and results).

  1. Statistical significance described as p should be found in every figure description and in all proper places in text, because authors use four different p values and is hard to remember which “set of stars” describes which p value.

Response: We thank you for your comments and suggestions regarding the p values. We have now specified the p values in result description and in figure legends in the revised manuscript.

  1. Figure 4: in panel C, authors described third blot image as Smad2, but in description there is a sFRP-1 protein – please correct the description on the panel.

Response: We apologize for the error and thank you for pointing this out. Figure 4C has been corrected in the revised manuscript.

  1. Line 347: not “oil” but fat droplet.

Response: Thank you for pointing this out. We have changed “oil” to “fat droplet”.

  1. Methods section: not “humanely killed” but sacrificed or euthanized should be used in description of experiments with animals involved.

Response: Thank you for your comments. “sacrificed” has now been used in the revised manuscript.

  1. Line 405: “cells growing monolayer” means that some cells from rat’s bones were cultured? If there were cultured cells from bones after methotrexate treatment, they should not be mixed with cells and bone samples harvested after euthanasia, but prepared and examined separately and described in results.

Response: We apologize for confusing descriptions. The “cells growing monolayer” referred to the cultured cells, including MC3T3.E1 cells and 3T3 F442A cells. We have now clarified this in revised the manuscript.

Reviewer 2 Report

Authors in this study has demonstrated a role of miR-542-3p in regulating os-528 teogenesis and adipogenesis following MTX treatment. Presented results suggest that supplementation of miR-542-3p may attenuate MTX-542 induced impairment in bone formation and increase in bone marrow fat formation via 543 preserving Wnt/β-catenin signalling and TGF-β signalling following MTX treatment. Articleis well writen and reads good, some minor corections shuld be made, specially:

  1. The introduction not provide sufficient background and don't include all relevant references to the role of methotrexate treatment. More atention should be paid about the mechanism of influence on BMSCs cells,
  2. Why Materials and Methods are after results section, for the reader it should be first of all presentation of used methods than the obtained results.
  3. miRNA isolation kit, Total RNA Purification Kit - kit number should be presented
  4. Conclusions should be expanded and try to clarify the exact mechanisms of action not only syntetic presentation of results.

Author Response

We highly appreciated your expert insightful and constructive comments and suggestions for this work. The manuscript has been revised carefully according to the comments and suggestions. Changes are shown as red color texts in the revised manuscript. Please find below our responses to your specific comments:

  1. The introduction not provide sufficient background and don't include all relevant references to the role of methotrexate treatment. More atention should be paid about the mechanism of influence on BMSCs cells.

Response: We appreciated your comments. In Introduction in the revised manuscript, we have now provided more relevant information and references on the effect of methotrexate (MTX) treatment in BMSC cells.

  1. Why Materials and Methods are after results section, for the reader it should be first of all presentation of used methods than the obtained results

Response: Thank you for your comment. This manuscript was prepared using the IJMS template and thus the list of sections cannot be modified.

  1. miRNA isolation kit, total RNA Purification Kit - kit number should be presented.

Response: Thank you for your comments. The kit has been described in the revised manuscript as GenEluteTM Total RNA Purification Kit (RNB100-100RXN, Sigma-Aldrich, NSW, Australia).

  1. Conclusions should be expanded and try to clarify the exact mechanisms of action not only syntetic presentation of results.

Response: We appreciate your advice. We have now revised Conclusions in the revised manuscript.